# The mitochondrial copper chaperone COX11 has an additional role in cellular redox homeostasis

Ivan Radin[1¤]*, Luise Kost[1], Uta Gey[1]*, Iris Steinebrunner[2], Gerhard Rödel[1]*

1 Institute for Genetics, Technische Universität Dresden, Dresden, Germany, 2 Department of Biology, Technische Universität Dresden, Dresden, Germany

¤ Current address: Department of Biology, Washington University in St. Louis, St. Louis, Missouri, United States of America

* iradin@wustl.edu (IR); uta.gey@tu-dresden.de (UG); gerhard.roedel@tu-dresden.de (GR)

**Data Availability Statement:** All relevant data are within the manuscript and its Supporting Information files.

**Funding:** This work was supported by a PhD fellowship to IR from the Dresden International

## Abstract

Mitochondria are sites of cellular respiration, which is accompanied by the generation of dangerous reactive oxygen species (ROS). Cells have multiple mechanisms to mitigate the dangers of ROS. Here we investigate the involvement of the COX complex assembly chaperone COX11 (cytochrome *c* oxidase 11) in cellular redox homeostasis, using homologs from the flowering plant *Arabidopsis thaliana* (AtCOX11) and yeast *Saccharomyces cerevisiae* (ScCOX11). We found that *AtCOX11* is upregulated in Arabidopsis seedlings in response to various oxidative stresses, suggesting a defensive role. In line with this, the overexpression of either *AtCOX11* or *ScCOX11* reduced ROS levels in yeast cells exposed to the oxidative stressor paraquat. Under normal growth conditions, both Arabidopsis and yeast COX11 overexpressing cells had the same ROS levels as the corresponding WT. In contrast, the COX11 knock-down and knock-out in Arabidopsis and yeast, respectively, significantly reduced ROS levels. In yeast cells, the ScCOX11 appears to be functionally redundant with superoxide dismutase 1 (ScSOD1), a superoxide detoxifying enzyme. The ΔSccox11ΔScsod1 mutants had dramatically reduced growth on paraquat, compared with the WT or single mutants. This growth retardation does not seem to be linked to the status of the COX complex and cellular respiration. Overexpression of putatively soluble COX11 variants substantially improved the resistance of yeast cells to the ROS inducer menadione. This shows that COX11 proteins can provide antioxidative protection likely independently from their COX assembly function. The conserved Cys219 (in AtCOX11) and Cys208 (in ScCOX11) are important for this function. Altogether, these results suggest that COX11 homologs, in addition to participating in COX complex assembly, have a distinct and evolutionary conserved role in protecting cells during heightened oxidative stress.

## Introduction

For many organisms, aerobic cellular respiration is an essential process, which converts chemical energy stored in sugars and other metabolites into ATP. This complex process is completed

Graduate School for Biomedicine and Bioengineering (DIGS-BB) (https://www.digs-bb.de), which is funded by the DFG (German Research Foundation). The Publication Fund of the Technische Universität Dresden provided funding for open access publishing of this paper. The funders had no role in study design, data collection and analysis, decision to publish, or preparation of the manuscript.

**Competing interests:** The authors have declared that no competing interests exist.

by the mitochondrial electron transport chain (ETC) which shuttles electrons from NAD(P)H and succinate to the terminal acceptor, molecular oxygen [1]. During this process some electrons escape and reduce molecular oxygen, generating superoxide, which can subsequently be converted into other reactive oxygen species (ROS) [2]. While respiratory complexes represent a major source of ROS in mitochondria, several other redox reactions also contribute to ROS production [3]. It is estimated that 1–5% of molecular oxygen is converted to ROS [4].

ROS molecules are highly reactive and can oxidize and thereby damage other molecules such as lipids, proteins, and nucleic acids. Consequently, organisms have evolved complex mechanisms to control ROS levels and reduce their toxicity and detrimental effects (reviewed in [2] and [3]). Some of them are well characterised, for example, the enzyme family of superoxide dismutases (SOD), which convert superoxide ions into oxygen and hydrogen peroxide [5]. The contribution of other proteins to oxidative metabolism is less well understood. One such example is the COX11 (cytochrome *c* oxidase 11) protein family.

Based on data mainly obtained from studies in yeast and bacteria, it is assumed that the main role of COX11 proteins is to deliver $Cu^+$ to the $Cu_B$ centre of the COX1 subunit of the COX complex (cytochrome *c* oxidase or Complex IV of the respiratory chain) [6]. Dimeric COX11 proteins [7] are present in most respiring organisms, from which the homolog of the yeast *Saccharomyces cerevisiae* (ScCOX11) is probably the best-studied family member [8–10]. In our previous work, we identified and characterised the *Arabidopsis thaliana* COX11 homolog (AtCOX11) [11]. This protein was, like the yeast counterpart, localised to mitochondria, presumably to the inner membrane, and involved in COX complex assembly. Interestingly, not only knockdown (KD) but also overexpression (OE) of *AtCOX11* reduced COX complex activity by ~50% and ~20%, respectively [11]. We proposed that both surplus and shortage of COX11 may interfere with the fine-tuned copper delivery balance necessary for COX complex assembly. In line with this, the absence of ScCOX11 leads to a non-functional COX complex and respiratory deficiency in yeast [6, 8, 12].

However, members of this conserved protein family might also be involved in mitochondrial oxidative metabolism [13–16]. Pungartnik *et al.* [13] showed that the yeast *Sccox11* null mutant is highly sensitive to the ROS-inducing chemicals N-nitrosodiethylamine and 8-hydroxyquinoline. Subsequently, Khalimonchuk *et al.* [14] and Veniamin *et al.* [15] demonstrated that the Δ*Sccox11* strain also showed an increased sensitivity to hydrogen peroxide when compared with the WT strain. The direct scavenging of ROS was also suggested for rice (*Oryza sativa*) COX11 homologue (OsCOX11) [16]. The authors reported that OsCOX11 dysfunction leads to a loss of pollen viability, presumably because the timing of a ROS burst necessary for pollen maturation is disturbed. In this work, we present further evidence that supports the proposed involvement of COX11 proteins in cellular redox homeostasis.

## Results

### Oxidative stress induces *AtCOX11* expression

We analysed the promoter region of *AtCOX11* for the presence of *cis*-active ROS-responsive elements which are prevalent in known ROS-induced genes [17, 18]. In *AtCOX11*, the noncoding region upstream of the start codon harbours as many as 16 putative oxidative-stress-responsive elements (Fig 1A and S1 Fig). In contrast, the promoter region of *AtHCC1* (homolog of copper chaperone SCO1 (synthesis of cytochrome *c* oxidase 1)), another mitochondrial chaperone that delivers copper to the COX complex [19], contains only five ROS-responsive consensus sequences (S1 Fig).

The overrepresentation of putative ROS-responsive elements prompted us to analyse the expression levels of *AtCOX11* transcripts under oxidative stress (Fig 1B). We treated WT

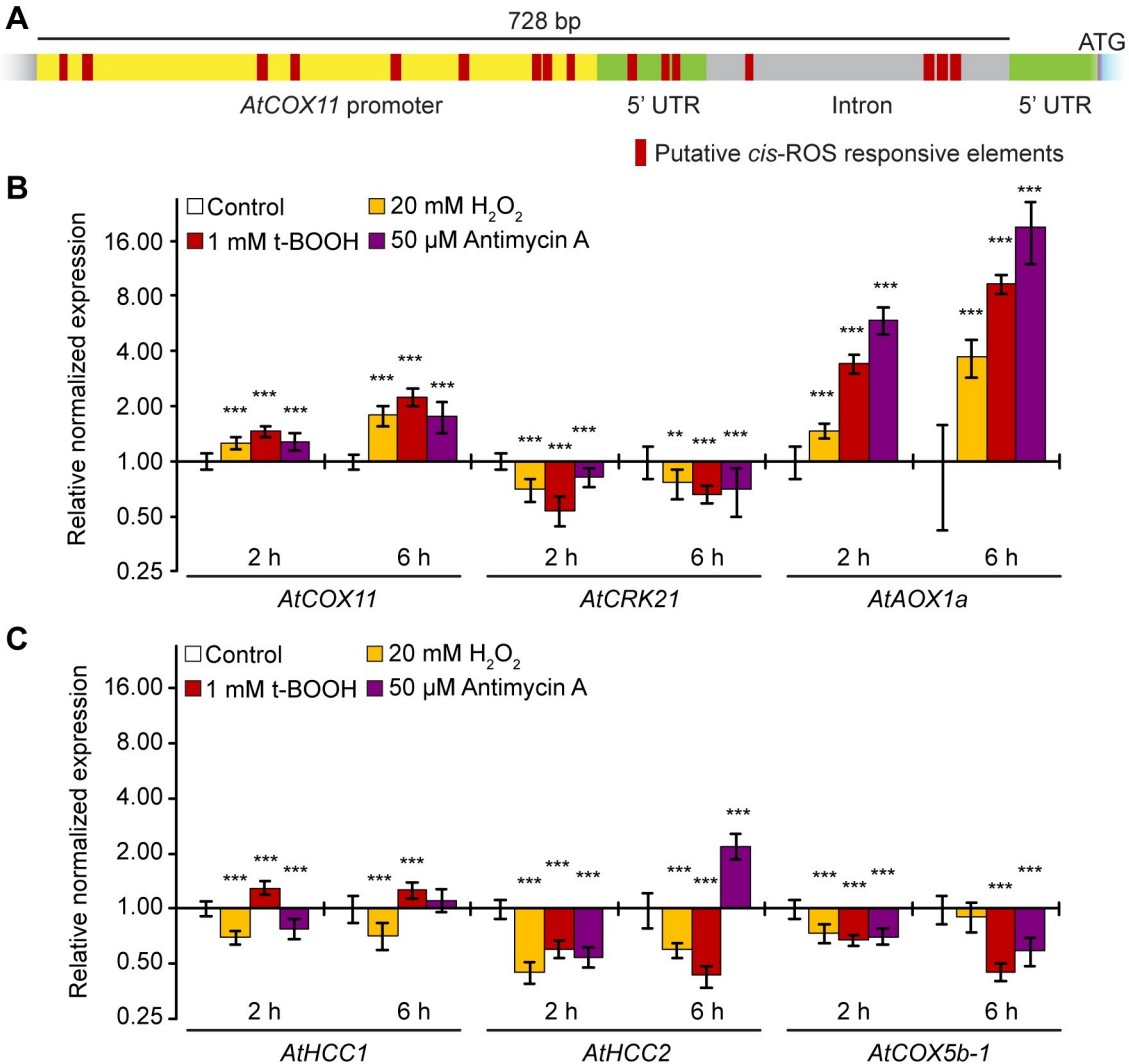

**Fig 1. *AtCOX11* is upregulated by oxidative stress. (A)** Scaled diagram of putative ROS-responsive elements within the *AtCOX11* promoter and 5'UTR. **(B)** and **(C)** Gene regulation in response to oxidative stress. Stress was applied for 2 h and 6 h. Mean values of mRNA levels in treated samples were normalized to the control sample and plotted on a logarithmic scale (base 2). Values and statistical significance compared with the control sample (**P < 0.01; ***P < 0.001) were calculated with the CFX manager software, Bio-Rad (unpaired Student's t-test). Error bars represent ± standard deviation (SD). Exact values and SD are listed in the S1 Table.

seedlings for 2 h or 6 h with the commonly used concentrations of the oxidative reagents hydrogen peroxide ($H_2O_2$, 20 mM) [20], tert-Butyl hydroperoxide (t-BOOH, 1 mM) [21], and antimycin A (50 µM) [22] followed by qPCR analysis. Hydrogen peroxide is a ROS molecule that easily transverses membranes and induces oxidative stress throughout the whole cell, while the organic peroxide t-BOOH is transported to mitochondria as well as other cellular compartments [23]. Antimycin A induces ROS production at the mitochondrial ETC by inhibiting the respiratory complex III [24].

*AtCRK21* (<u>c</u>ysteine-<u>r</u>ich receptor-like protein <u>k</u>inase <u>21</u>) and *AtAOX1a* (<u>a</u>lternative <u>ox</u>idase <u>1a</u>) are known to be down- and upregulated by ROS [25], respectively, and were chosen as controls. As expected, oxidative stress reduced *AtCRK21* levels, while *AtAOX1a* mRNA abundance was increased about 19-fold (Fig 1B).

*AtCOX11* was slightly upregulated (~1.3 fold) in response to all three 2-h oxidative stress conditions. The upregulation further increased to ~2 fold after 6 h. These data suggest that at least some of the regulatory elements present in the *AtCOX11* promoter region are functional.

To check the specificity of the *AtCOX11* ROS-response profile, we analysed the expression of other genes coding for COX complex subunit and assembly factors. The transcript levels of the copper chaperone *AtHCC1* were only marginally affected at both time points (Fig 1C). On the other hand, its homolog *AtHCC2* (homolog of copper chaperone SCO2), which lacks a copper-binding motif [26], was affected by ROS. It showed a 50% decrease in the transcript level (Fig 1C). The *AtHCC2* promoter region carries seven putative ROS-responsive elements (S1 Fig). *AtHCC2* levels, 2.2 times higher after a 6-h antimycin A treatment, were a notable exception from the otherwise observed downregulation.

The transcript levels of another COX-related gene, the COX complex subunit *AtCOX5b-1*, were reduced by ~30% after 2 h of oxidative stress and by ~50% after 6 h, except for the $H_2O_2$ treatment, which had no effect at this time point (Fig 1C). Clearly, not all mitochondrial COX complex-related genes respond to ROS in the same way. Our qPCR analysis yielded similar results as previously observed in large-scale microarray experiments [22] (Genevestigator database) (S2 Table).

Taken together, the increased transcript abundance of *AtCOX11* in response to ROS may hint at some role in mitochondrial ROS homeostasis.

## Knockdown of *AtCOX11* reduces cellular ROS

To explore the role of AtCOX11 in redox homeostasis further, ROS levels were measured in the Arabidopsis *COX11* knock-down (KD) and overexpression (OE) plant lines that were generated previously [11]. The *AtCOX11* mRNA levels in KD plants were approximately 30% of the WT levels, while in the two overexpression lines OE1 and OE2, the transcript amounts were approximately 6- and 4-fold higher, respectively [11]. Cellular oxidative status or ROS levels were measured by two independent methods: indirectly by determining the lipid peroxidation levels (Fig 2A), and directly by staining isolated protoplasts with the ROS-specific dye DCFDA (2′,7′ dichlorofluorescin diacetate) (Fig 2B).

For lipid peroxidation measurements, plants were grown for 14 h in the dark prior to the experiments to minimize ROS contributions from photosystems. Then, the leaves were harvested to measure malondialdehyde (MDA) and hydroxyalkenals (HAE) concentrations, typical products generated by decomposing lipid peroxides.

MDA and HAE levels were lower in all KD lines compared with the WT, albeit only statistically significant for KD1-1 and KD1-2 plants (Fig 2A). The levels in the OE lines were indistinguishable from the WT, possibly because of relatively low *AtCOX11* overexpressing levels (~5-fold increase [11]).

This was confirmed by a second assay, in which protoplasts were incubated with the DCFDA dye, which upon entering the cell and oxidation exhibits a bright green fluorescence. All KD lines showed significantly lower cellular ROS levels compared with the WT and again, the OE lines were indistinguishable from the WT (Fig 2B). Of note is that these assays detect ROS from the entire cell and might not be sensitive enough to detect subtle changes within the intermembrane space (IMS) of mitochondria.

These results seemingly contradict a function of Arabidopsis COX11 in ROS defense. However, the observed phenotypes in the KD lines could be related to the loss of COX complex activity (see discussion for details). In summary, two different ROS detection methods revealed a reduction in ROS levels when *AtCOX11* expression was reduced, but no change in ROS amounts when *AtCOX11* was overexpressed.

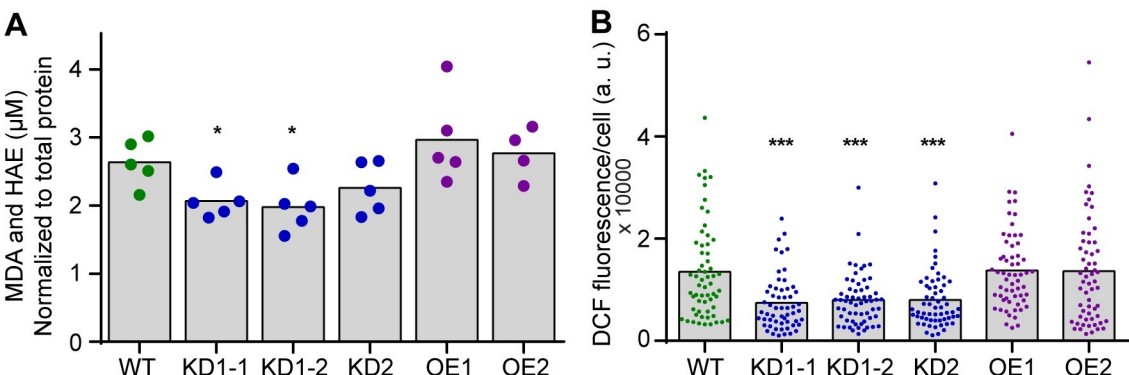

**Fig 2. Disturbance of *COX11* expression in Arabidopsis alters cellular ROS levels. (A)** Lipid peroxidation was determined in *AtCOX11* knock-down (KD) and overexpression (OE) mutant plant lines by measuring the concentration of malondialdehyde (MDA) and hydroxyalkenals (HAE) (normalized to the total protein) in the leaves. Each dot corresponds to an independent experiment. **(B)** Total fluorescence of protoplasts, isolated from WT and *AtCOX11* KD and OE mutant plant lines and stained with DCFDA, in arbitrary units (a. u.). Each dot corresponds to an individual protoplast. Data is an amalgamation of three independent experiments. Asterisks indicate statistically significant difference (unpaired Student's t-test; *P < 0.05, ***P < 0.001) between mutants and WT. The average values and standard deviations are given in the S1 Table.

## COX11 proteins play a role in oxidative stress tolerance in yeast

Next, we investigated the role of COX11 proteins in ROS homeostasis in another model organism, the budding yeast. For this, *ScCOX11* was knocked out or overexpressed (alternatively *AtCOX11)* and the effects on cellular oxidative status or ROS levels were studied under normal and acute oxidative stress conditions (Fig 3A and 3B). We choose paraquat (PQ) as an oxidative stressor because it's known to induce superoxide production in mitochondria [27]. Yeast cells were stained with the ROS-specific dye DCFDA, and the green fluorescence of each cell was measured by flow cytometry. For each data set, the mode (defined as a number that occurs most often in the data set) was determined. As such, mode corresponds to the X-axis position of the peak of the cell fluorescence intensity histogram (S2 Fig). Modes from three independent experiments were averaged and depicted as bar graphs (Fig 3A and 3B). Cell fluorescence intensity distributions from individual experiments are depicted in S2 Fig.

Similar to what we observed in Arabidopsis (Fig 2), the Δ*Sccox11* strain, which has a non-functional COX complex, showed a significant reduction in the cellular ROS levels compared with the WT strain (Fig 3A). When the cells were treated with 2 mM PQ for 30 min prior to DCFDA staining, the WT had significantly higher ROS levels, while the Δ*Sccox11* had similar levels compared with the untreated controls (Fig 3A).

Under normal growth conditions, the *Sc/AtCOX11* overexpressing yeast strains had similar ROS levels as the empty vector control (Fig 3B). However, when the same strains were pre-treated with 2 mM PQ, the increase of ROS levels in COX11 overexpressing yeast cells was significantly lower than in the corresponding empty vector control (Fig 3B). This indicates that the overexpression of *COX11* genes can partly alleviate the imposed oxidative stress. These differences in cellular ROS levels did not affect the growth of yeast strains under PQ stress (S3A Fig). We also tested the growth of *COX11* overexpressing strains on respiratory YPE (ethanol) media, which was indistinguishable from the WT. (S3B Fig). This suggests that *COX11* overexpression does not meaningfully affect cellular respiration.

An intriguing possibility might be that the role of ScCOX11 in ROS defense is redundant with main ROS defense mechanisms such as the action of ScSOD1, which is localized both in the cytoplasm and mitochondrial IMS [28]. To test this hypothesis, we generated a *ScCOX11* and *ScSOD1* double-deletion mutant (Δ*Sccox11*Δ*Scsod1*) and compared its growth under

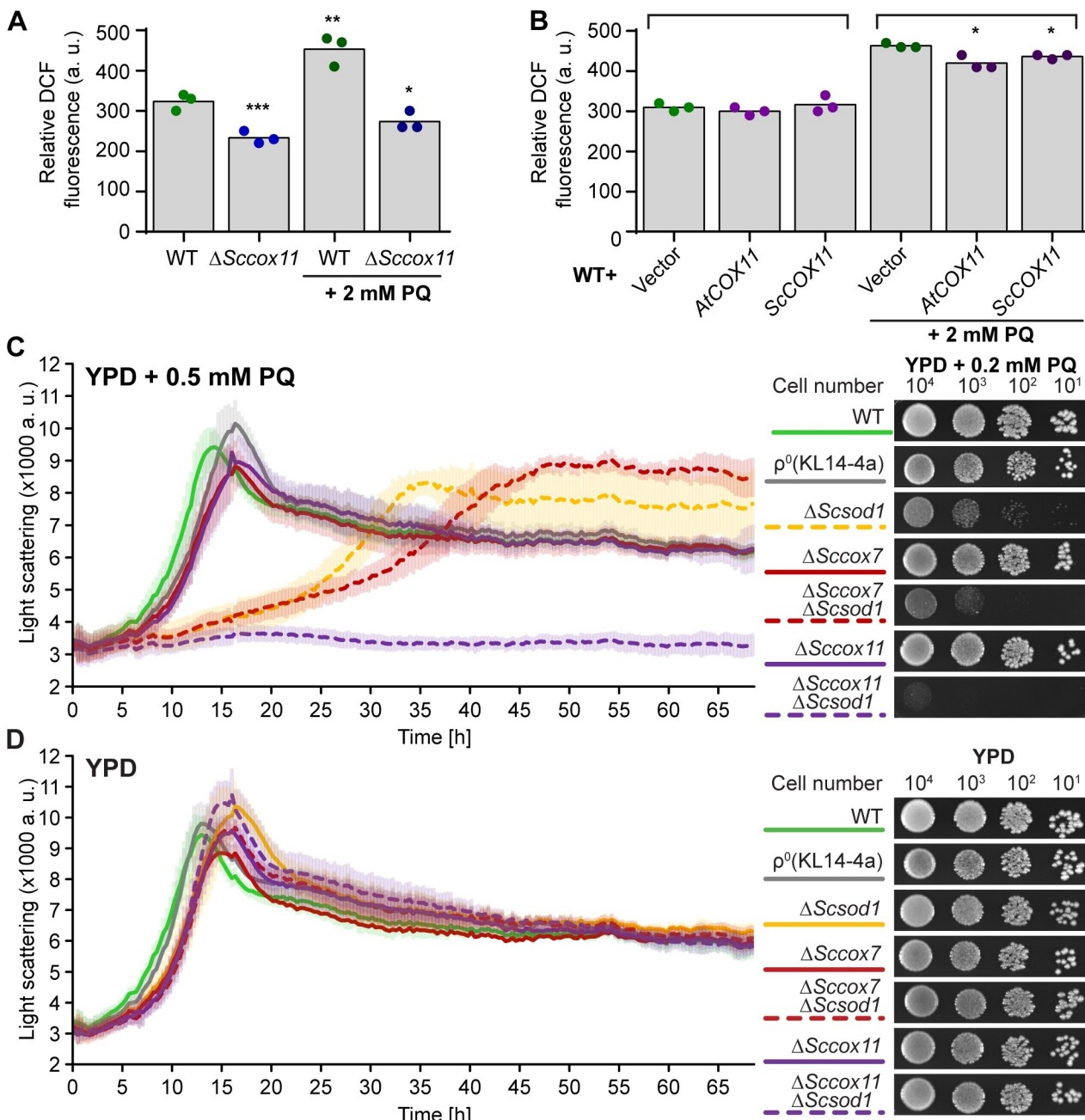

**Fig 3. COX11 proteins influence yeast oxidative stress tolerance. (A)** and **(B)** ROS levels determined by DCFDA staining of WT, Δ*Sccox11* (A), and *AtCOX11* or *ScCOX11* overexpressing (B) yeast strains; after either mock treatment or treatment with 2 mM paraquat (PQ). The total fluorescence of individual cells was measured by flow cytometry. Descriptive statistics for the cytometry datasets are given in the S1 Table. Each dot represents the result of an independent experiment, while the bars represent the averages. Asterisks indicate statistically significant difference (unpaired Student's t-test; *P < 0.05, **P < 0.01, ***P < 0.001) between mode averages of mutant strains compared with the untreated WT (A) or between empty vector control (treated or untreated) and COX11 overexpressing strains from the same treatment (B). **(C)** and **(D)** Growth of the indicated yeast deletions strains in liquid (left panel) and on solid (right panel) YPD media in the presence (C) or absence (D) of PQ. The growth of yeast strain in liquid media was measured with Nephelometry, in 3 independent experiments (with three technical replicates each). Error bars are ± SD (standard deviation). On the right are example images of the dilution series of the same strains spotted on solid media and cultured for 48 h.

continuous PQ stress to WT and corresponding single mutants. For these assays, we chose a PQ concentration that would not affect the growth of the WT (Fig 3C and 3D). For yeast liquid growth assay, we used a slightly higher PQ concentration, because under those growth conditions yeast cells exhibit higher oxidative stress tolerance. The ΔScsod1 strain had moderate growth retardation in the presence of PQ, compared with the WT and ΔSccox11, which were similar (Fig 3C). Under the same conditions, the ΔSccox11ΔScsod1 double mutant was severally affected in its growth, both in liquid and on solid media. However, it's important to note that ΔSccox11 yeast has a non-functional COX complex, and hence is respiratory deficient (S3C Fig), which could also affect mitochondrial ROS metabolism. To test this possibility, we included a $\rho^0$ strain, which lacks the entire mitochondrial genome and ETC, as well as respiratory deficient ΔSccox7 single and ΔSccox7ΔScsod1 double mutant strains (S3C Fig). ScCOX7 is a structural COX complex subunit required for its function [29]. Both $\rho^0$ and ΔSccox7 strains had the same growth efficiency on PQ as ΔSccox11, while the ΔSccox7ΔScsod1 growth was only slightly slower than ΔScsod1 (Fig 3C). This shows that the loss of the COX complex and respiratory deficiency on their own do not explain the observed PQ hypersensitivity of the ΔSccox11ΔScsod1 strain. We confirmed this further by testing the PQ sensitivity of the ΔSccyt1ΔScsod1 yeast strain, which lacks Cytochrome c1 from the third respiratory (cytochrome b) complex and is also respiratory deficient (S3C Fig) [30]. Similar to the ΔSccox7ΔScsod1 results, the growth of the ΔSccyt1ΔScsod1 strain was comparable to the ΔScsod1 single mutant (S3D Fig). All strains had comparable growth under normal control conditions (liquid or solid YPD media without PQ) (Fig 3D and S3D Fig), excluding the possibility that knockouts affected the general growth fitness.

Taken together, these data suggest that ScCOX11 has an additional function in ROS defence, which is partially redundant with ScSOD1.

## Truncated COX11 proteins improve yeast growth under oxidative stress

The functional, copper-binding domain of COX11 proteins is localized in the mitochondrial IMS [9, 10], which is a rather small compartment. The contributions of other larger cellular compartments to the ROS production and metabolism might mask COX11-induced changes within the IMS. In order to get around this problem and test whether COX11 proteins can provide oxidative protection independently of their natural environment, we generated truncated putatively soluble protein versions (tCOX11) (Fig 4A and S4 Fig). These versions lacked both their mitochondria targeting signal and transmembrane domain, hence should localize to the cytosol. Furthermore, we wanted to test a possible mechanism of action for COX11 antioxidative activity. Mature COX11 proteins have three highly conserved cysteines (Fig 4A and S4 Fig) [7], which could be oxidized by ROS and form inter- or intramolecular disulphide bridges. To test their importance, we generated tAt/ScCOX11 protein variants where either all three (ΔCys) or individual cysteines were mutated to alanines. This gave us a set of proteins with the ability to form varying number of disulphide bridges (Fig 4, diagrams on the right-hand side).

Using immunoblotting and ScCOX11 specific antibody [8], we tested stability and expression levels of tScCOX11 protein variants in WT yeast cells. Most variants (tScCOX11, C111A, C208A, and C210A) were expressed to a comparable level, except for tScCOX11 ΔCys, which appears to be unstable and/or degraded (Fig 4B). In addition to the main band of tScCOX11 monomers (expected size 22.6 kDa), we also observed a higher band that could correspond to SDS- and β- mercaptoethanol-resistant COX11 dimers or oligomers, which we observed before [11]. The antibody was not sensitive enough to detect the endogenous ScCOX11, which is present at a very low abundance in the total protein extracts. Additionally, as this antibody

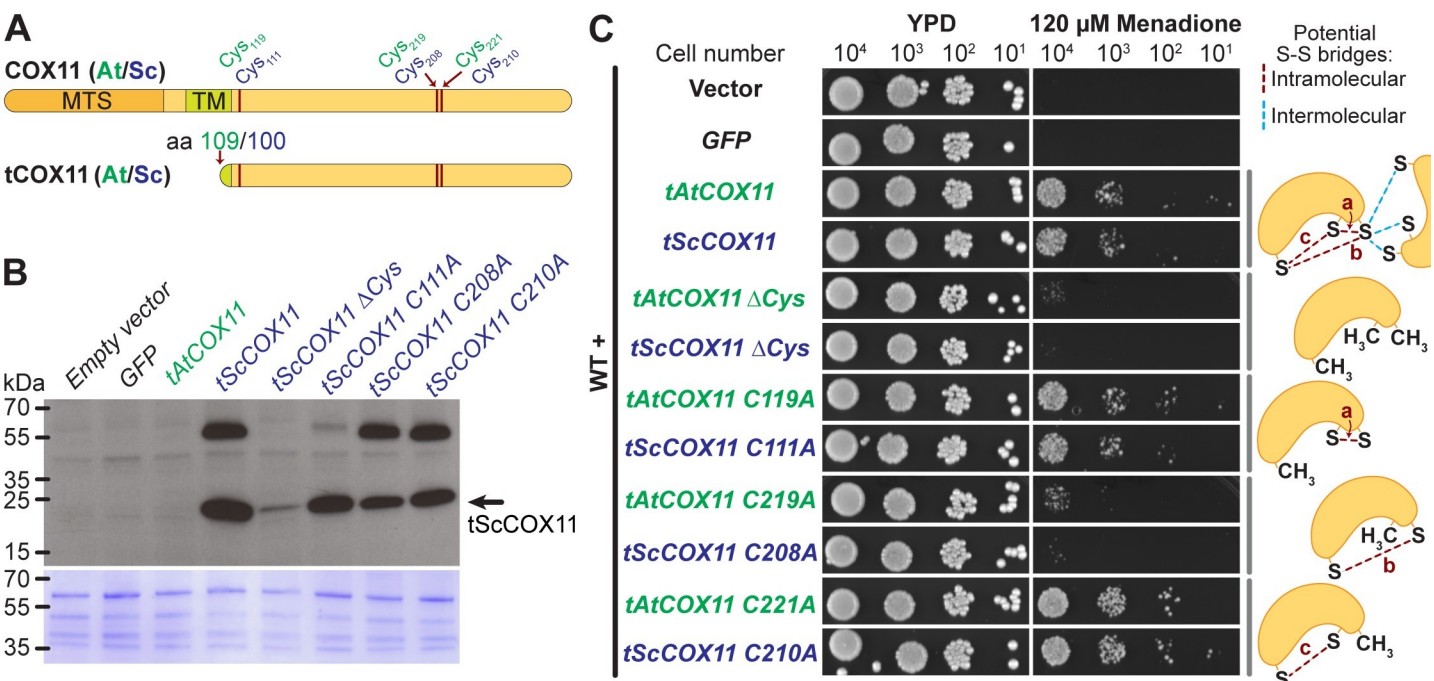

**Fig 4. Truncated COX11 proteins improve yeast growth under oxidative stress. (A)** Scaled diagram of full-length COX11 proteins from *Arabidopsis thaliana* and *Saccharomyces cerevisiae* (top scheme) and truncated versions (tAtCOX11 missing amino acids (aa) 2–108 and tScSCOX11 missing aa 2–100) (bottom scheme). The positions of conserved cysteines (Cys) are indicated. MTS (mitochondrial targeting signal), TM (transmembrane domain). **(B)** Immunoblot with total protein extracts from WT yeast strains expressing free GFP or tCOX11 variants (top panel). Blot was probed with ScCOX11 specific antibody. Bottom panel, Coomassie staining of the same blot. The original images are given in S1 Raw images. **(C)** The growth of WT yeast overexpressing different versions of Arabidopsis or yeast tCOX11 proteins under oxidative stress. Serial dilutions of yeast strains were spotted on YPD media with or without oxidative stressor menadione. In the mutated tCOX11variants, the conserved cysteines were replaced with alanines. Experiments were repeated at least 3 times with similar results. Diagrams on the right visualize possible intramolecular (red dashed lines) and some intermolecular (blue dashed lines) disulphide-bridges.

was specifically raised against yeast homologs, we were, unfortunately, not able to detect the tAtCOX11 variants (Fig 4B).

To test the ability of the truncated COX11 proteins to confer oxidative stress resistance to yeast cells, we overexpressed tAtCOX11, tScCOX11, and their cysteine-mutated variants in WT cells and tested their growth in the presence or absence of oxidative stress. For this assay, we chose to use menadione which is a general redox cycler and ROS inducer [31], as the tCOX11 variants were expressed in the cytosol. As negative controls, WT cells were also transformed with either empty vector or vector expressing free GFP which is of similar size to tCOX11 and without any know antioxidative activity. All strains grew normally on control YPD media, ruling out negative effects of protein overexpression (Fig 4C, left panels). In the presence of 120 μM menadione, control strains (with empty and GFP-expressing vector) failed to grow, as did the *tAtCOX11 C219A*, *tScCOX11 C208A*, and both *ΔCys* strains. On the other hand, *tAtCOX11*, its *C119A/C221A* mutants, *tScCOX11* and its *C111A/C210A* mutants were able to grow quite efficiently (Fig 4C right panels).

This shows that COX11 proteins (tAt/ScCOX11) are able to increase the oxidative stress tolerance of yeast cells, even when expressed in a non-native compartment. The $Cys_{219}$ and $Cys_{208}$ in AtCOX11 and ScCOX11 respectively, appear to be important for that function.

## Discussion

The role of COX11 proteins as copper chaperones in COX complex assembly has been well documented [8, 9, 11, 12, 32], however, an additional antioxidative role has also been shown

[13–16]. Here we explore this further and present additional evidence of COX11 protein's participation in cellular redox homeostasis.

The analysis of Arabidopsis *COX11* gene response to cellular oxidative stress showed a pattern of gene upregulation (Fig 1B), in contrast to other mitochondrial genes related to the COX complex or its assembly, which showed either minimal change (*AtHCC1*) or downregulation (*AtHCC2* and *AtCOX5b-1*) (Fig 1C). This could suggest that higher levels of AtCOX11 are required by the plant cells to deal with the imposed oxidative stress. However, it is well known that changes in mRNA levels do not always correlate with changes in protein levels, so further studies will be needed to investigate the physiological relevance of the observed upregulation. Nevertheless, the notion that cells need higher amount of COX11 proteins to deal with imposed oxidative stress is supported by the observation that overexpression of either *AtCOX11* or *ScCOX11* diminished the PQ-induced ROS levels in yeast cells (Fig 3B). On the other hand, under normal growth conditions, the overexpression of COX11 proteins did not alter cellular ROS levels neither in Arabidopsis (Fig 2) nor yeast (Fig 3B). This suggests that COX11 plays a role in redox homeostasis only under heightened oxidative stress. Alternatively, the ROS changes within mitochondrial IMS during normal growth are small and get masked by other larger cellular compartments. Another caveat worth mentioning is that the overexpression of other cysteine-containing proteins could potentially convey similar oxidative stress protection as observed with COX11 proteins.

When the COX11 function was disrupted, either through gene knock-down in Arabidopsis or knock-out in yeast, the overall cellular ROS levels were significantly reduced (Figs 2 and 3A). The exact reason for this is not clear, but it is possibly linked to the COX complex deficiency caused by the disruption of COX11. Some studies [33, 34], show that COX complex deficiency leads to reduced ROS levels, in agreement with our findings, while others show higher ROS levels [35]. Other studies further still show that mitochondria of COX mutants are not a major source of ROS [36, 37], but in some cases can promote ROS production by other cellular compartments [36]. As such loss of COX complex activity can affect cellular ROS levels, but the exact mechanisms still need to be elucidated.

Further evidence that COX11 proteins are involved in mitochondrial redox homeostasis comes from the *ScCOX11* and *ScSOD1* double-deletion yeast strains (Fig 3C). The double mutant had a severely diminished growth under oxidative conditions, compared with the WT or either single mutant. This suggests that ScCOX11, and presumably other COX11 homologs, could work in tandem with SOD1 to maintain redox homeostasis in the IMS, either by directly affecting ROS levels or in some way modulating the activity of SOD1. Alternatively, ScCOX11 could serve as a backup mechanism when a major antioxidant system, like ScSOD1 fails, which increases the cellular ROS levels [38]. This heightened oxidative stress could shift COX11 proteins, from their normal function in COX biogenesis into a new role in antioxidative defence. As mentioned before, the loss of ScCOX11 also leads to the loss of COX complex activity. However, it appears that COX deficiency and respiratory deficiency, in general, do not change the resistance of cells to PQ-induced oxidative stress. The Δ*Sccox7*Δ*Scsod1* and Δ*Sccyt1*Δ*Scsod1* strains, which are respiratory deficient due to the lack of the IV (COX) and III respiratory complexes, respectively, grew similar to Δ*Scsod1* single mutant in the presence of PQ (Fig 3C and S3C, S3D Fig). The $\rho^0$ strain, which lacks the whole ETC, as well as Δ*Sccox11*, Δ*Sccox7*, *and* Δ*Sccyt1* single mutants grew similar to the WT, both in the presence and absence of PQ (Fig 3C, 3D and S3D Fig). This suggests that the severe growth retardation of Δ*Sccox11*Δ*Scsod1* is specific to ScCOX11 and not to a general loss of COX complex or respiratory competence.

Nevertheless, defects in COX complex and respiratory activity affect many aspects of cellular function, so it is hard to completely separate COX11's copper chaperone role and any

potential antioxidative activity. To this end, we generated truncated versions of AtCOX11 and ScCOX11 (tCOX11) for cytosolic expression (Fig 4A). This allowed us to test the antioxidative activity of COX11 in a different subcellular compartment separate from its respiratory function. Western blot analysis, using ScCOX11 specific antibody, confirmed that tScCOX11 is stably expressed (Fig 4B), while the absence of growth defects on YPD media (Fig 4C, left panels), ruled out substantial toxicity of tCOX11 proteins. Overexpression of either tAtCOX11 or tScCOX11 permitted the yeast cells to grow in the presence of ROS-inducing menadione (Fig 4C, right panels), showing that COX11 proteins are indeed able to protect cells against the oxidative stress, independently of their COX complex assembly role. Yeast cells expressing soluble GFP failed to grow on menadione (Fig 4C, right panels), showing that the overexpression of a random protein does not confer oxidative stress tolerance.

How COX11 proteins protect the cells from oxidative stress, is still an open question. One possibility is that COX11 proteins scavenge ROS directly by the formation of inter- or intramolecular disulphide bridges (S-S) between the conserved cysteines (Fig 4 and S4 Fig) as previously reported for other ROS protectants, e.g. the human PRX3 (peroxiredoxin-3) [3]. To test this, we individually mutated three conserved cysteines, which did not affect the protein stability (Fig 4B) and found that Cys219 (in *A.t.*) / Cys208 (in *S.c.*) is important for the antioxidant role of COX11 (Fig 4C), in agreement with previous work [15]. This cysteine could, upon oxidation by ROS molecules, form intramolecular disulphide bridges (labelled as "a" and "c" on diagrams in Fig 4C) with either of the two remaining cysteines. Bode *et al.* [39] provided experimental evidence that two cysteines within the copper-binding motif (S4 Fig) could form a disulphide bridge (bridge "a" in Fig 4C). On the other hand, the disulphide bond prediction software DiANNA 1.1 predicted only the formation of bridge "c", albeit with a low probability score (Arabidopsis/yeast COX11 bridge "a": 0.01/0.01, bridge "b": 0.01/0.01 and bridge "c": 0.12/0.16; maximum score: 1). The conserved cysteines could also form intermolecular S-S bridges between the two COX11 subunits or between COX11 and other proteins. Of note is that based on bacterial COX11 (*Sinorhizobium meliloti*) crystal structure [40] and human COX11 model [41], all three cysteines are on the protein surface and thus easily accessible to oxidation by ROS molecules. After their oxidation by ROS molecules, these putative disulphide bridges could be resolved by other proteins or the oxidized COX11s could be degraded as a non-functional proteins.

The formation of a disulphide bridge is not the only possible explanation for the antioxidant activity of COX11 proteins. As copper chaperones, they could potentially use the bound metal ions to detoxify ROS. However, the C221A (*A.t.*) and C210A (*S.c.*) mutants still exhibit antioxidative activity (Fig 4C), despite lacking the copper-binding motif. We also cannot exclude the possibility that overexpression of tCOX11 affects menadione uptake in the yeast cells. However, this seems unlikely if we consider that cells expressing C219A (*A.t.*) and C208A (*S.c.*) mutant variants are susceptible to menadione toxicity.

Unfortunately, as the ScCOX11 antibody does not recognize the Arabidopsis homologs, we cannot be certain that tAtCOX11 variants are stable in yeast cells. However, based on the identical growth patterns on menadione (Fig 4C), we can surmise that tAtCOX11 variants behave similar to their yeast counterparts. It is intriguing that we observed very similar results with two evolutionarily distant homologs, suggesting that role in redox homeostasis is conserved and possibly a fundamental property of all COX11 proteins. Previously, we reported that AtCOX11 cannot functionally complement the respiratory deficiency of Δ*Sccox11* [11], likely because specific protein-protein interactions are required (e.g. COX11 and COX1 or COX11 and COX17 [7]). On the other hand, the AtCOX11, like ScCOX11, can confer protection to the yeast cells against oxidative stress in some situations. This means that the role of COX11 proteins in redox homeostasis likely does not require the involvement of other proteins.

While many open questions remain regarding the role of COX11 proteins in redox homeostasis, the data presented here show that the Arabidopsis and *S. cerevisiae* COX11 homologs can relieve oxidative stress in cells, possibly using their conserved cysteines. The COX11 proteins, as chaperones, are in close proximity to the ROS-generating respiratory complexes and are perfectly poised to quickly detoxify ROS and prevent damage. Alternatively, they could be recruited as a second line of antioxidative defence in situations of elevated mitochondrial oxidative stress or when the primary defence mechanisms fail.

## Material and methods

### Plant material and culture conditions

*Arabidopsis thaliana* (*At*) Columbia (Col) 0 was used as the WT. The *AtCOX11* knock-down (KD) and overexpressing (OE) lines were previously generated and characterised [11]. KD1/OE lines and KD2 lines were used in T3 and T2 generations, respectively.

Plants were grown either on MS (1x Murashige and Skoog salts, 1% [w/v] sucrose, 0.5 g/L 4-morpholineethanesulfonic acid [MES], 0.8% [w/v] agar) plates or on soil (Einheitserde, type P, Pätzer, Sinntal-Jossa, Germany; mixed with sand 4:1, fertilised by watering with 0.1% [v/v]) Wuxal Basis, Aglukon). For protoplast generation, the MS + 1% sucrose media (for KD lines) was supplemented with 30 μg/mL of kanamycin.

Plants were cultured in a growth chamber with a light intensity of 150 μmol/m$^2$s, relative humidity of 35%, and day/night temperatures of 24/21°C, respectively. Two types of day/night cycles were used: long day (16-h d) and short day (10-h d).

### Yeast material and culture conditions

*Saccharomyces cerevisiae* (*Sc*) WT strain BY4741 (Accession (Acc.) number (no.) Y00000) and deletion strains Δ*Sccox11* (Acc. no. Y06479), Δ*Sccox7* (Acc. no. Y00842), Δ*Scsod1* (Acc. no. Y06913 and Y16913), and Δ*cyt1* (Acc. no. Y11841) were obtained from EUROSCARF (Frankfurt, Germany). The Δ*cox11*Δ*sod1* (MAT *a*; *his3Δ1; leu2Δ0; ura3Δ0; YJR104c::kanMX4; YPL132w::kanMX4*), Δ*cox7*Δ*sod1* (MAT *α*; *his3Δ1; leu2Δ0; met15Δ; ura3Δ0; YJR104c::kanMX4; YMR256c::kanMX4*), and Δ*cyt1*Δ*sod1* (MAT *α*; *his3Δ1; leu2Δ0; lys2Δ0; ura3Δ0; YJR104c::kanMX4; YOR065w::kanMX4*) were generated by crossing the respective single-deletion strains followed by sporulation, tetrad dissection and analysis. Rho$^0$ strain (KL14-4a) [42], which lacks mitochondria DNA was used as a control.

Constructs used for *COX11* overexpression (*pAG415ADH-AtCOX11* and *pAG415ADH-Sc-COX11*) were generated previously [11]. To create the cytosolic versions of COX11, fragments were amplified by PCR (for primer sequences and cloning details see S3 Table) and inserted by Gateway cloning into pDONR or pENTR vectors. All constructs were moved into the high-copy yeast expression-vector pAG425GPD-ccdB-EGFP [43]. Yeast cells were transformed as described in Gietz and Schiestl [44]. Transformed yeast strains were cultured on minimal media (0.5% [w/v] ammonium sulphate, 0.19% [w/v] yeast nitrogen bases, 2% [w/v] glucose, 2.5% [w/v] agar and required amino acids). For liquid cultures, yeast strains were cultured at 30°C with shaking at 180 rpm.

### Yeast growth assays

For growth analysis on solid media, vector transformed yeast strains were cultured in liquid minimal media, while knock-out strains were cultured in liquid YPD for 24 h, then diluted with minimal media to OD$_{600}$ = 0.05 and cultured for another 16 h. Serial dilutions were spotted on solid YPD media plates YPD (yeast peptone dextrose) media (1% [w/v] yeast extract,

2% [w/v] peptone, 2% [w/v] glucose, 2% [w/v] agar). Appropriate oxidative stressors (paraquat (methyl viologen) and menadione) were added after autoclaving, from freshly prepared stocks to the YPD media (cooled to 55°C). Oxidative stress plates were used within 24 h. Growth was documented after incubation for 48–60 h at 30°C.

For growth analysis by Nephelometry, cells were incubated in liquid YPD for 24 h, followed by 1:100 dilution in fresh media and overnight incubation. For each well, $10^4$ cells were used to inoculate 200 µl of the YPD media with or without 0.5 mM PQ. For each genotype and condition, three technical replicates were inoculated. Growth analysis was performed in clear flat-bottom 96-well plates at 30°C with constant shaking, and light scattering was measured in 20-min intervals using the NEPHELOstar (BMG Labtech, Ortenberg, Germany) [45].

To test their respiratory competence, serial dilutions of yeast cells were spotted on YP (1% [w/v] yeast extract, 2% [w/v] peptone) media supplemented with 3% (v/v) ethanol (YPE) or 2% (v/v) glycerol (YPG) or both 3% (v/v) ethanol (YPE) and 2% (v/v) glycerol (YPEG), and cultured for 48–72 h.

## Bioinformatic analysis

Arabidopsis and yeast gene and protein sequences were obtained from The Arabidopsis Information Resource [46] and the GenBank [47], respectively. For protein sequence alignment, the EMBOSS Needle software (The European Bioinformatics Institute) [48] was used. For the prediction of targeting signal cleavage sites, TargetP [49, 50] was used, and the transmembrane domains were predicted with TMHMM2.0 [51]. Disulphide bridge formation was predicted with DiANNA 1.1 [52–54]. The Genevestigator was used to examine public microarray databases [55].

## Stress treatments and qPCR

For the oxidative stress treatments, the Arabidopsis WT seedlings were cultured on solid MS plates + 1% (w/v) sucrose for 12 days. Stress was applied for 2 h or 6 h by placing seedlings on the surface of liquid MS + 1% (w/v) sucrose media supplemented with the appropriate stressor. Antimycin A (Sigma Aldrich) was dissolved in absolute ethanol and subsequently diluted with MS media. As a control, seedlings were placed on the surface of liquid MS + 1% (w/v) sucrose media without the stressors. Immediately after the stress treatment, the seedlings were frozen in liquid nitrogen, and RNA was isolated. RNA isolation and quantitative real-time RT-PCR (qPCR) were performed as previously described [11]. The RNA quality was analysed with the BioAnalyzer 2100 (Agilent, USA), and only RNAs with RNA integrity numbers (RIN) in the range of 7.5 to 8.5 were reverse transcribed. The efficiency and optimal concentrations of all primer pairs were experimentally determined and are listed in the S4 Table. The data were statistically analysed (unpaired Student's t-test) with the Bio-Rad CFX Manager 3.1 software.

## Lipid peroxidation measurement

The levels of lipid peroxidation were determined with the Bioxytech LPO-586 kit (OxisResearch). Rosette leaves from plants (10 weeks old) grown under short-day conditions were harvested at the beginning of the light period and immediately ground with a pestle in a mortar with 500 µL of grinding buffer (20 mM Tris-Cl pH 7.4, 5 mM butylated hydroxytoluene) per 100 mg of tissue. The leaf suspension was cleared by two centrifugation steps (each 3,000g for 10 min at 4°C). Of the final supernatant, 7 µL and 100 µL were used for quantitation of protein concentration (Bio-Rad DC assay) and lipid peroxidation measurements, respectively. The "reagent 2" (methanesulfonic acid) was employed to determine the amounts of malondialdehyde (MDA) and 4-hydroxyalkenals (HAE). All samples were run in triplicates and readout

with a TECAN M200 plate reader (Tecan). The lipid peroxidation levels were normalised to the protein concentrations in the supernatants.

## ROS level measurement in protoplasts

Protoplasts were isolated as previously described [56] with slight modifications. Of the protoplasting buffer (20 mM KCl, 20 mM MES, 0.4 M mannitol, 1.25% [w/v] cellulase R-10, 0.3% [w/v] macerozyme R-10, 10 mM $CaCl_2$, 0.1% [w/v] BSA, pH 5.7) 1.5 mL were added to approximately 100 mg of finely cut 12-d-old seedlings cultured under long-day conditions. After 4 h of agitation at room temperature, the suspensions were successively filtered through 100- and 50-μm meshes. Protoplasts were pelleted (280$g$ for 10 min at 4˚C) and washed first with W5 buffer (154 mM NaCl, 125 mM $CaCl_2$, 5 mM KCl, 5 mM MES, pH 5.7), and then with MMG buffer (0.4 M mannitol, 15 mM $MgCl_2$, 4 mM MES, pH 5.7). They were finally resuspended and stored in MMG buffer at 4˚C until use. All buffers were prepared fresh.

For determination of ROS levels, protoplasts were incubated with 5 μM DCFDA for 10 min [57] and then imaged with the LSM780 microscope from Zeiss (C-Apochromat 40x/1.20 W Korr M27 objective, excitation with a 488-nm laser, detection in 510–542 nm range for DCF (2′,7′ dichlorofluorescin) and 647–751 nm for chlorophyll autofluorescence). The total fluorescence for individual protoplasts was determined with the Fiji image analysis software [58] as raw integrated density in the green channel. Protoplasts with a large number of chloroplasts, which are derived from photosynthetic tissues, were excluded to avoid the measurement of chlorophyll autofluorescence and ROS produced by photosystems.

## ROS level measurement in yeast cells

Liquid YPD media was inoculated with the respective strains and cultured for 24 h. Then the cultures were diluted to $OD_{600}$ of 0.01, grown overnight (14–16 h), and used to start the final YPD cultures (starting $OD_{600}$ = 0.1), which were incubated at 30˚C until an $OD_{600}$ of 0.5–0.6 was reached. This successive refreshing was necessary to ensure the same physiological state of all strains. Cultures were aliquoted (1 mL each) into 2-mL tubes and either treated with water (= mock) or with 2 mM PQ for 30 min at 30˚C with agitation. Subsequently, cells were pelleted (3500$g$ for 3 min at RT) and washed twice with PBS. Finally, cells were resuspended in 1 mL of PBS and split into two aliquots. One was used as the negative control, while the other was stained with DCFDA (final concentration 20 μM) for 45 min at 30˚C with agitation. After staining, cells were washed twice with PBS. Total DCF fluorescence was measured in the CyFlow SL (Partec, Germany) with 488-nm excitation and detection in the FL1 channel (527 nm/BP 30 nm). FL1 channel gain was set to a level on which fluorescence could not be observed in negative unstained controls.

## Western blot

For total yeast protein isolation, overnight yeast cultures (in minimal selection media) were used to inoculate 10 mL YPD media (starting $OD_{600}$ = 0.5), which was cultured until $OD_{600}$ reached roughly 2. Yeast cells were spun down, washed once with water, resuspended in 500 μL of 20 mM TRIS-Cl pH 7.6 with proteinase inhibitor cocktail (Sigma-Aldrich), and mixed with 400 μL of glass beads (Ø 0.25–0.5 mm). The mixture was shaken on the Mixer mill MM200 (Retsch) for 5 min (30 swings per s), cooled on ice, and centrifuged (15000 g for 1 min at 4˚C). The supernatant was used as a total protein extract and protein concertation was determined with the DC Protein Assay from BioRad.

For SDS-PAGE, 10-well 4–15% gradient Mini-PROTEAN® TGX™ Precast Gels from Bio-Rad were used. For each well, 25 μg of total protein was mixed with loading buffer (final concentration: 50 mM TRIS-Cl pH 6.8, 2% (w/v) SDS, 0.1% (w/v) Bromophenol blue, 10% (v/v)

glycerol, and 1% (v/v) β- mercaptoethanol), incubated at 90˚C for 5 min, cooled down on ice, spun down or 1 min at 15000 g, and loaded onto the gel. The gel was run in 25 mM TRIS, 250 mM Glycine, and 0.1% (w/v) SDS at 200 V for approximately 40 min. Subsequently, the proteins were transferred to the PVDF membrane using Wet Transfer and Mini Trans-Blot® Module from Bio-Rad (transfer buffer: 25 mM Tris, 192 mM glycine, 15% (v/v) methanol), for 1h at 100 V at 4˚C. The membrane was blocked overnight in in TBS-T (100 mM Tris, 150 mM NaCl, 0.1% (v/v) Tween 20, pH with HCl to 7.5) + 5% (w/v) non-fat milk powder at 4˚C, probed with αScCOX11 [8] antibody (1:2500 dilution in TBS-T + 5% (w/v) non-fat milk powder) for 1 h at room temperature, washed with TBS-T, incubated with secondary antibody (1:10000 αChicken-HRP (5220–0373 from SeraCare) in TBS-T + 5% (w/v) non-fat milk powder) for 30 min at room temperature, washed again with TBS-T, and detected with Super-Signal™ West Dura Substrate (Thermo Scientific) and X-ray films. Original films are given in S1 Raw images. After the detection, proteins on the PVDF were stained with Coomassie [59] for 5 min in 50% (v/v) methanol, 7% (v/v) acetic acid and 0.1% (w/v) Coomassie Blue R, followed by 5 min distaining in 50% (v/v) methanol and 7% (v/v) acetic acid, and rinse in 90% (v/v) methanol and 10% (v/v) acetic acid.

## Supporting information

**S1 Raw images. Original western blot films.**
(PDF)

**S1 Fig. Cis-acting putative ROS-response elements in the putative promoter regions of AtCOX11, AtHCC1, and AtHCC2.**
(PDF)

**S2 Fig.** Total cell fluorescence intensity distributions of *ScCOX11* knock-out (A, B, C) and overexpressing yeast (D, E, F) cells stained with DCFDA.
(PDF)

**S3 Fig. Extended data for Fig 3.**
(PDF)

**S4 Fig. Alignment of *A. thaliana* (*A. t.*) and *S. cerevisiae* (*S. c.*) COX11 protein sequences.**
(PDF)

**S1 Table. Absolute and normalized values from bar graphs and descriptive statistics.**
(XLSX)

**S2 Table. Gene regulation in response to 50 μM antimycin A (microarray data from Ng *et al.* [2013], Genevestigator).**
(PDF)

**S3 Table. Cloning primers.**
(PDF)

**S4 Table. Primers used for qPCR.**
(PDF)

## Acknowledgments

We thank Oleh Khalimonchuk for supplying us with αScCOX11 antibody, and EUROSCARF (Germany) for yeast strains. We are very grateful to Elizabeth S. Haswell for supporting Ivan Radin, to complete the necessary experiments for this publication.

## Author Contributions

**Conceptualization:** Ivan Radin, Uta Gey, Iris Steinebrunner, Gerhard Rödel.

**Data curation:** Ivan Radin, Iris Steinebrunner, Gerhard Rödel.

**Formal analysis:** Ivan Radin, Luise Kost.

**Funding acquisition:** Ivan Radin, Uta Gey, Gerhard Rödel.

**Investigation:** Ivan Radin, Luise Kost.

**Methodology:** Ivan Radin, Luise Kost, Uta Gey.

**Project administration:** Ivan Radin, Iris Steinebrunner, Gerhard Rödel.

**Resources:** Ivan Radin, Iris Steinebrunner, Gerhard Rödel.

**Supervision:** Uta Gey, Iris Steinebrunner, Gerhard Rödel.

**Validation:** Ivan Radin, Luise Kost, Uta Gey, Iris Steinebrunner, Gerhard Rödel.

**Visualization:** Ivan Radin.

**Writing – original draft:** Ivan Radin, Luise Kost, Uta Gey, Iris Steinebrunner, Gerhard Rödel.

**Writing – review & editing:** Ivan Radin, Luise Kost, Uta Gey, Iris Steinebrunner, Gerhard Rödel.

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
