## [Decision Letter · Decision Letter 0]

7 Jul 2021

PONE-D-21-15368

The mitochondrial copper chaperone COX11 has an additional role in cellular redox homeostasis

PLOS ONE

Dear Dr. Radin,

Thank you for submitting your manuscript to PLOS ONE. After careful consideration, we feel that it has merit but does not fully meet PLOS ONE’s publication criteria as it currently stands. Therefore, we invite you to submit a revised version of the manuscript that addresses the points raised during the review process.

Specifically, the reviewers recommend extensive editing of the text to fix mistakes, recognize limitations and provide a more in-depth critical analysis of the results. The reviewer's also highlight the need of additional experiments and controls to fully support the author's conclusions. It is essential that you address all the concerns listed below

A rebuttal letter that responds to each point raised by the academic editor and reviewer(s). You should upload this letter as a separate file labeled 'Response to Reviewers'.A marked-up copy of your manuscript that highlights changes made to the original version. You should upload this as a separate file labeled 'Revised Manuscript with Track Changes'.An unmarked version of your revised paper without tracked changes. You should upload this as a separate file labeled 'Manuscript'

We look forward to receiving your revised manuscript.

Kind regards,

Rodrigo Franco

Academic Editor

PLOS ONE

Journal Requirements:

In your cover letter, please note whether your blot/gel image data are in Supporting Information or posted at a public data repository, provide the repository URL if relevant, and provide specific details as to which raw blot/gel images, if any, are not available. Email us at plosone@plos.org if you have any questions

Reviewers' comments:

Reviewer's Responses to Questions

**Comments to the Author**

1. Is the manuscript technically sound, and do the data support the conclusions?

Reviewer #1: Partly

Reviewer #2: Yes

Reviewer #3: No

2. Has the statistical analysis been performed appropriately and rigorously? 

Reviewer #1: Yes

Reviewer #2: Yes

Reviewer #3: I Don't Know

3. Have the authors made all data underlying the findings in their manuscript fully available?

Reviewer #1: No

Reviewer #2: Yes

Reviewer #3: Yes

4. Is the manuscript presented in an intelligible fashion and written in standard English?

Reviewer #1: Yes

Reviewer #2: Yes

Reviewer #3: Yes

5. Review Comments to the Author

Reviewer #1: Cox11 is a conserved assembly factor of cytochrome oxidase. It is a protein of the intermembrane space that is attached to the inner membrane by a membrane anchor. Three cysteine residues are conserved. Previous studies suggested that Cox11 supports the copper insertion into subunit 1 (Cox1) in a redox-dependent reaction. It’s function in this reaction might be similar to that of the better characterized Sco1/Sco2 protein in metalation of subunit 2 (Cox2).

This study consists of two loosely connected parts, none of which is really developed very much. The first part (Figs 1&2) comes to the conclusion that the expression of Cox11 in Arabidopsis is induced by highly unphysiological amounts (20 mM!) of hydrogen peroxide, and that Cox11 mutants show somewhat lower ‘ROS’. The authors conclude that Cox11 is a redox enzyme, however, we know from transcriptome and genome-wide mutation studies that hundreds of genes show similar phenotypes, many of which are certainly no ‘redox enzymes’.

The second part is an analysis of yeast mutants which express variants of the yeast and Arabidopsis Cox11. This part is more interesting and the data in this part are more compelling. These data suggest that cysteines in Cox11 are relevant for menadione-resistance and for the formation of Cox11 dimers.

In summary, the function of Cox11 is unresolved. Thus, this manuscript addresses an interesting question. The yeast data are of good quality, even though they do not go much beyond previous studies of the Glerum, Winge and Herman labs. Still, these data nicely confirm the relevance of cysteine residues in Cox11 and will be of some interest for specialists.

Specific points

1. The authors should avoid the umbrella term ‘ROS’ and try to be chemically correct. Superoxide dismutase does not remove ROS, but converts the ROS superoxide into the ROS hydrogen peroxide. I recommend reading some recent literature on oxidative stress, e.g. Sies et al, Annu Rev Biochem (PMID: 28441057). I guess, the conclusion here is that the loss of Cox11 leads to an increase of the levels of superoxide in the intermembrane space.

2. Mutants in cytochrome oxidase lead to increased superoxide production (see for example Dubinski et al., PMID: 29567354; Khalimonchuk et al., PMID: 19995914). The observed synthetic defect of COX11 and SOD1 mutants is easily explained by this. The speculations of COX11 being an SOD-like enzyme should therefore be down-graded, as evidence is not presented here.

3. The authors propose, that Cox11 is – in addition to its function in cytochrome oxidase biosynthesis – a redox enzymes. The authors should repeat their experiments in rho0 cells (which lack cytochrome oxidase). Only if they also there observe an protective effect of Cox11 on redox resistance, this claim should be made. Even though the authors describe this strategy in the text, I did not see convincing data in the figures. Without further experimental support, this speculation has to be removed from the study. Sentences like ‘Taken together, these data suggest that ScCOX11 has an additional function separate from its COX complex assembly role, in ROS defence, which is partially redundant with ScSOD1.’ should be avoided! They are not based on compelling evidence.

4. The authors conclude from Figure 1: ‘Taken together, AtCOX11 showed a unique ROS response, and the increased transcript abundance could suggest that AtCOX11 plays some role in mitochondrial ROS homeostasis.’ However, since they just tested the expression of six genes, they cannot conclude that COX11 has a unique expression signature! Either they use a transcriptome-wide approach or, simpler, remove this bold statement.

5. Fig. 3C lacks a control plate without paraquat. This needs to be added. In general, it is no good style to use spliced figures. Since the authors have to repeat this central experiment anyway, they should drop all strains side-by-side onto the same plates and avoid splicing.

Reviewer #2: This work aimed to reveal the COX11’s function in cellular redox homeostasis in addition to its known role in cytochrome oxidase assembly. The authors adequately addressed most of the comments and suggestions of previous reviewers. The data provide some valuable information for a better understanding of COX11’s functions and action mechanisms. For instance, it’s intriguing that C208A site-directed substitution, but not C210A, in tScCOX11, leads to its functional defect in menadione resistance.

Despite the sound approaches and interesting observations, the physiological relevance of the finding remains unclear. There is no evidence indicating that the oxidant-induced up-regulation of COX11 observed in plants is relevant to its roles in redox homeostasis/antioxidant defense. Artificial over-expression of any thioredoxin-like proteins may confer resistance to oxidative stress. The authors could point out these limitations of the presented work in the Discussion section. Secondly, the authors could consider both intramolecular and intermolecular disulfide bonds in predicting the potential S-S bridges in Fig. 4C. Thirdly, in Fig. 4B, the evidence supporting that the ~55 kDa bands are Cox11 dimers is unclear. The authors should describe details about SDS-PAGE conditions. Do the bands disappear on reducing gels? Lastly, this manuscript does not contain figure legends, which is problematic in reviewing it.

Reviewer #3: In this manuscript, the authors have focused on elucidating the redox role of a mitochondrial copper chaperone, Cox11. This work was motivated by previous studies, which have shown that in addition to copper delivery to mitochondrial cytochrome c oxidase, Cox11 plays a role in defense against external oxidants. The authors used two model systems – Arabidopsis thaliana and the budding yeast, Saccharomyces cerevisiae to suggest evolutionarily conserved cytoprotective role of Cox11 during heightened oxidative stress. Specifically, the authors used Cox11 knockdown and overexpression mutants of A. thaliana and S. cerevisiae and tested their ability to counteract different oxidants. Uncovering non-mitochondrial role of Cox11 is interesting but works appears preliminary and has numerous weaknesses.

Major concerns:

1. Lack of novelty. As the authors themselves pointed out in the introduction, previous studies by Winge group and Glerum group have already shown increased sensitivity of S. cerevisiae Cox11 mutants to hydrogen peroxide.

2. Some of the results are internally inconsistent and counter-intuitive. For example, data from figure 1 and figure 2 are paradoxical. If the role of Cox11 is to counter against oxidative stress, how come COX11 knockdown lines have reduced markers of oxidative stress? The authors attribute this observation to the role of Cox11 in mitochondrial electron transport chain function. If so, the authors need to perform control experiments using chemical or genetic inhibitors of the electron transport chain.

3. There is no consistency in using oxidant or their concentrations. For example, in Figure 1, the authors used hydrogen peroxide and tert-butyl hydroperoxide, Figure, 3 paraquat and Figure 4, menadione. It is not clear why? Similarly it is not clear as to why in Figure 3, the authors have used 2 mM PQ to measure ROS, 0.5 mM PQ to measure growth in liquid media and 0.2 mM PQ for growth in solid media.

4. The western blotting results in Fig 4B where the authors show the expression of Cox11 variants in yeast WT cells raises concerns because it is not clear as to why a band for endogenous WT Cox11 protein is not seen in all samples.

5. The biological relevance of the results obtained using putative cytosol localized truncated Cox11 forms is questionable. WT Cox11 is an inner mitochondrial membrane protein and unlike some of the other IMS soluble proteins, it is neither dual localized nor can it shuttle between cytosol and mitochondria.

6. The use of DCFDA to measure cellular ROS should be complemented with MitoSOX to measure mitochondrial ROS in various truncation versions as well in mitochondrial localized Cox11.

7. The data in Fig 3C right panel contradicts published literature on the sensitivity of yeast cox11∆ to oxidative stressors. See: “Khalimonchuk O, Bird A, Winge DR. Evidence for a pro-oxidant intermediate in the assembly of cytochrome oxidase. J Biol Chem. 2007 Jun 15;282(24):17442-9.”

Minor concerns:

1. The authors should provide information about the statistical tests used. (Ex: Fig 1B and C).

2. There are minor corrections required: change ScSOX11 to ScCOX11 in lines 286 and 287.

6. PLOS authors have the option to publish the peer review history of their article (what does this mean?). If published, this will include your full peer review and any attached files.

Reviewer #1: No

Reviewer #2: No

Reviewer #3: No

---

## [Author Response · Author response to Decision Letter 0]

20 Aug 2021

We thank all the reviewers for their comments and suggestions to improve this manuscript. 

PONE-D-21-15368

The mitochondrial copper chaperone COX11 has an additional role in cellular redox homeostasis

PLOS ONE

Comments to the Author

1. Is the manuscript technically sound, and do the data support the conclusions?

Reviewer #1: Partly

Reviewer #2: Yes

Reviewer #3: No

 2. Has the statistical analysis been performed appropriately and rigorously? 

Reviewer #1: Yes

Reviewer #2: Yes

Reviewer #3: I Don't Know

We have added the missing information to the Figure 1 legend and the methods section. The qPCR data from Figure 1 were analyzed with the Bio-Rad CFX Manager 3.1 software. The unpaired Student’s t-test was used.

3. Have the authors made all data underlying the findings in their manuscript fully available?

Reviewer #1: No

Reviewer #2: Yes

Reviewer #3: Yes

It is not obvious to us which data are not available. In case data are missing, we will provide them.

 4. Is the manuscript presented in an intelligible fashion and written in standard English?

 Reviewer #1: Yes

Reviewer #2: Yes

Reviewer #3: Yes

 5. Review Comments to the Author

Reviewer #1: Cox11 is a conserved assembly factor of cytochrome oxidase. It is a protein of the intermembrane space that is attached to the inner membrane by a membrane anchor. Three cysteine residues are conserved. Previous studies suggested that Cox11 supports the copper insertion into subunit 1 (Cox1) in a redox-dependent reaction. It’s function in this reaction might be similar to that of the better characterized Sco1/Sco2 protein in metalation of subunit 2 (Cox2).

This study consists of two loosely connected parts, none of which is really developed very much. The first part (Figs 1&2) comes to the conclusion that the expression of Cox11 in Arabidopsis is induced by highly unphysiological amounts (20 mM!) of hydrogen peroxide, and that Cox11 mutants show somewhat lower ‘ROS’. 

For our short-term acute oxidative stress of Arabidopsis seedlings, we used relatively high concentrations of oxidative stressors that have been used previously in the literature. The use of hydrogen peroxide in 10-20 mM range or even higher is a common practice in Arabidopsis research (for example: DOI: 10.1104/pp.127.1.159, 10.1105/tpc.111.090894, 10.1074/jbc.M605293200). We have added new references to the text to demonstrate the previous use of the same concentrations of the stressors. 

The authors conclude that Cox11 is a redox enzyme, however, we know from transcriptome and genome-wide mutation studies that hundreds of genes show similar phenotypes, many of which are certainly no ‘redox enzymes’.

Could the reviewer please specify in which line we state that Cox11 is a redox enzyme? We tried not to make such a claim as currently there is not sufficient evidence for that. In this version of the manuscript, we have also added to the discussion, the idea that COX11 proteins might be degraded after oxidation by ROS and not recycled like enzymes. We discuss other possible non-enzymatic explanations for the observed antioxidative role of COX11 as well. In addition, could the reviewer please clarify what transcriptome and genome-wide mutation studies and phenotypes they are referring to? In case the comment was in connection to our interpretation of Arabidopsis qPCR results (Fig. 1) we modified the text and limited our claims to COX complex-related genes, which are the most relevant for the interpretation of the COX11 results. 

The second part is an analysis of yeast mutants which express variants of the yeast and Arabidopsis Cox11. This part is more interesting and the data in this part are more compelling. These data suggest that cysteines in Cox11 are relevant for menadione-resistance and for the formation of Cox11 dimers.

In summary, the function of Cox11 is unresolved. Thus, this manuscript addresses an interesting question. The yeast data are of good quality, even though they do not go much beyond previous studies of the Glerum, Winge and Herman labs. Still, these data nicely confirm the relevance of cysteine residues in Cox11 and will be of some interest for specialists.

Specific points

1. The authors should avoid the umbrella term ‘ROS’ and try to be chemically correct. Superoxide dismutase does not remove ROS, but converts the ROS superoxide into the ROS hydrogen peroxide. I recommend reading some recent literature on oxidative stress, e.g. Sies et al, Annu Rev Biochem (PMID: 28441057). I guess, the conclusion here is that the loss of Cox11 leads to an increase of the levels of superoxide in the intermembrane space.

We purposely chose to use general terms like ROS, because the current data does not allow for distinction between different types of ROS molecules. Both lipid peroxidation and DCFDA measure the general redox status of the cell. The reviewer rightfully pointed out that SOD1 converts superoxide into water and hydrogen peroxide, which we stated in the introduction. In the abstract, we did refer to SOD1 as a ROS detoxifying enzyme, which we now changed to avoid confusion. 

2. Mutants in cytochrome oxidase lead to increased superoxide production (see for example Dubinski et al., PMID: 29567354; Khalimonchuk et al., PMID: 19995914). The observed synthetic defect of COX11 and SOD1 mutants is easily explained by this. The speculations of COX11 being an SOD-like enzyme should therefore be down-graded, as evidence is not presented here.

Could the reviewer please clarify to which section of the suggested publications they were referring to? The Khalimonchuk et al. paper doesn’t mention superoxide or ROS while Dubinski et al. conclude: “Our results suggest that mitochondria may not be a major source of reactive oxygen species at stationary phase in cells lacking an intact respiratory chain.” We are aware that part of the literature reports increased ROS levels after disruption of COX complex, however, our data (Fig. 2, Fig. 3A) and other papers (doi:10.3390/cells9081843, 10.1073/pnas.0705738104) show the opposite effect. So it seems that the relationships between COX and respiration defects and ROS production are complex and possibly very dependent on the specific experimental model and setup. We have added this discussion to the manuscript. 

Regardless, this valid point was also raised in the first round of the reviews. To test this, we added the Δcox7 and Δcox7Δsod1 yeast strains to our experiments. These two strains, which are respiratory incompetent due to lack of the COX complex, grew similarly to WT and Δsod1, respectively, when exposed to PQ. This shows that the observed hypersensitivity of Δcox11Δsod1 stain to PQ is not linked to COX complex activity. Now, we also added an experiment with Δcyt1 and Δcyt1Δsod1 yeast strains which lack III respiratory complex and are also respiratory deficient (S3C, D Fig.). The Δcyt1Δsod1 strain grew the same on PQ as did the Δsod1, again confirming that respiratory deficiency on its own does not explain the PQ hypersensitivity of Δcox11Δsod1 yeast cells. Based on that we conclude that COX11 and SOD1 have an additive and possibly redundant function under these conditions, but we agree that at this time we can not conclude that COX11 is a SOD1-like enzyme. 

3. The authors propose, that Cox11 is – in addition to its function in cytochrome oxidase biosynthesis – a redox enzymes. The authors should repeat their experiments in rho0 cells (which lack cytochrome oxidase). Only if they also there observe an protective effect of Cox11 on redox resistance, this claim should be made. Even though the authors describe this strategy in the text, I did not see convincing data in the figures. Without further experimental support, this speculation has to be removed from the study. Sentences like ‘Taken together, these data suggest that ScCOX11 has an additional function separate from its COX complex assembly role, in ROS defence, which is partially redundant with ScSOD1.’ should be avoided! They are not based on compelling evidence.

We based our conclusion that the proposed antioxidative function, which is not necessarily an enzymatic function, of COX11 proteins is independent of their role in COX assembly, on the data with Δcox7Δsod1 strains as outlined above. However, the data from Figure 4 offer stronger support for this conclusion. There we show that truncated and cytosolic COX11 proteins can increase the oxidative stress tolerance of yeast cells. These data show that COX11 proteins have antioxidative properties even when present in a non-native compartment, the cytosol, where they don’t interact or affect the COX complex. We opted to use the cytosolic expression of COX11 over rho0 background, as the latter background has dramatic changes in many aspects on yeast physiology (including protein import, membrane potential, etc), making the interpretation of the experimental results much more challenging. 

Nevertheless, we have removed the conclusion that the antioxidative role of COX11 is separate from its COX chaperone role from the sentence the reviewer has mentioned because that sentence appears in the manuscript before all of the necessary data to support that conclusion are presented. 

4. The authors conclude from Figure 1: ‘Taken together, AtCOX11 showed a unique ROS response, and the increased transcript abundance could suggest that AtCOX11 plays some role in mitochondrial ROS homeostasis.’ However, since they just tested the expression of six genes, they cannot conclude that COX11 has a unique expression signature! Either they use a transcriptome-wide approach or, simpler, remove this bold statement.

We have ameliorated that overly broad statement.

5. Fig. 3C lacks a control plate without paraquat. This needs to be added. In general, it is no good style to use spliced figures. Since the authors have to repeat this central experiment anyway, they should drop all strains side-by-side onto the same plates and avoid splicing.

Those controls experiments were provided in Supplementary Fig 3. However, for clarity, we included the results in the main Figure 3D. 

Reviewer #2: This work aimed to reveal the COX11’s function in cellular redox homeostasis in addition to its known role in cytochrome oxidase assembly. The authors adequately addressed most of the comments and suggestions of previous reviewers. The data provide some valuable information for a better understanding of COX11’s functions and action mechanisms. For instance, it’s intriguing that C208A site-directed substitution, but not C210A, in tScCOX11, leads to its functional defect in menadione resistance.

Despite the sound approaches and interesting observations, the physiological relevance of the finding remains unclear. There is no evidence indicating that the oxidant-induced up-regulation of COX11 observed in plants is relevant to its roles in redox homeostasis/antioxidant defense. Artificial over-expression of any thioredoxin-like proteins may confer resistance to oxidative stress. The authors could point out these limitations of the presented work in the Discussion section. 

We have added these points to the discussion. 

Secondly, the authors could consider both intramolecular and intermolecular disulfide bonds in predicting the potential S-S bridges in Fig. 4C. 

We agree that this could also easily happen. We refer to it in the discussion and we have now added it to the graph in Fig. 4. 

Thirdly, in Fig. 4B, the evidence supporting that the ~55 kDa bands are Cox11 dimers is unclear. The authors should describe details about SDS-PAGE conditions. Do the bands disappear on reducing gels? 

The blot in Fig. 4B was from a standard reducing SDS page gel. We have added a more detailed Western blot protocol to the methods for clarification. With COX11 proteins, both yeast and Arabidopsis, we regularly observe SDS- and β-mercaptoethanol-resistant higher molecular weight bands (see doi: 10.3389/fpls.2015.01091), which by size could correspond to dimers or oligomers. However, we have not confirmed that experimentally, e.g., with protein mass spectrometry or similar methods, so we removed that from Fig 4, and only mention it in the text as a possible explanation. 

Lastly, this manuscript does not contain figure legends, which is problematic in reviewing it.

The figure legends were added to the main text after a paragraph where they were first mentioned as per PLOSone instructions for authors. 

Reviewer #3: In this manuscript, the authors have focused on elucidating the redox role of a mitochondrial copper chaperone, Cox11. This work was motivated by previous studies, which have shown that in addition to copper delivery to mitochondrial cytochrome c oxidase, Cox11 plays a role in defense against external oxidants. The authors used two model systems – Arabidopsis thaliana and the budding yeast, Saccharomyces cerevisiae to suggest evolutionarily conserved cytoprotective role of Cox11 during heightened oxidative stress. Specifically, the authors used Cox11 knockdown and overexpression mutants of A. thaliana and S. cerevisiae and tested their ability to counteract different oxidants. Uncovering non-mitochondrial role of Cox11 is interesting but works appears preliminary and has numerous weaknesses.

Major concerns:

1. Lack of novelty. As the authors themselves pointed out in the introduction, previous studies by Winge group and Glerum group have already shown increased sensitivity of S. cerevisiae Cox11 mutants to hydrogen peroxide. 

Those studies were thorough and very informative, however, they all used the same yeast strain and assay. Here, we confirmed their findings by using completely different yeast strains (different genetic backgrounds), oxidative stressors other than hydrogen peroxide, and different experimental setups. This not only strengthens their conclusions but also introduces novel assays for future work with COX11 and other mitochondrial mutants. 

In addition, we expanded their findings and show that COX11 proteins appear to be functionally redundant with SOD1 and that their putative antioxidative function is not directly linked to the COX complex or their localization to the mitochondria, so it is likely an intrinsic quality of COX11 proteins. And finally, we add an evolutionary component to the story by showing that the antioxidative functions appear to be conserved in phylogenetically very distant species (yeast and Arabidopsis), broadening the significance of the results to non-yeast researchers. 

2. Some of the results are internally inconsistent and counter-intuitive. For example, data from figure 1 and figure 2 are paradoxical. If the role of Cox11 is to counter against oxidative stress, how come COX11 knockdown lines have reduced markers of oxidative stress? The authors attribute this observation to the role of Cox11 in mitochondrial electron transport chain function. If so, the authors need to perform control experiments using chemical or genetic inhibitors of the electron transport chain.

We agree that this seems paradoxical, however, the literature shows that the relationship between COX defects and ROS production is very complex. Some publications show increased ROS productions after COX disruptions (doi: 10.1042/BJ20090214) while others show decreased ROS levels (doi:10.3390/cells9081843, 10.1073/pnas.0705738104), or even unchanged levels (doi: 10.3390/cells9020443). In other papers, it was shown that mitochondria of COX mutants are not a major source of ROS production (doi: 10.1016/j.bbabio.2018.03.011), however, they can influence other cellular compartments to produce ROS (doi: 10.1016/j.cmet.2013.07.005). So at this time, it is not clear why COX11 or other COX complex mutants have altered ROS levels. Nevertheless, it appears that depending on the model system (cell lines, plants, yeast strains) and experimental set-up different results regarding the effects of COX disruptions are obtained: ROS levels go up, down or remain unchanged. The scope of our paper is to show that COX11 has an effect on ROS levels in Arabidopsis and yeast and plays a role in the complex and intricate network of ROS regulation. The finding that COX11 knockdown lines have reduced markers of oxidative stress is indeed unexpected and therefore, more interesting and important to the field and future studies. For that reason, we opted to show that data in this manuscript.

We have added this discussion to the manuscript. We removed one possible explanation that lower ROS levels are due to the lower COX activity and now state that it is not clear why COX11 mutants have lower ROS levels. 

3. There is no consistency in using oxidant or their concentrations. For example, in Figure 1, the authors used hydrogen peroxide and tert-butyl hydroperoxide, Figure, 3 paraquat and Figure 4, menadione. It is not clear why? Similarly it is not clear as to why in Figure 3, the authors have used 2 mM PQ to measure ROS, 0.5 mM PQ to measure growth in liquid media and 0.2 mM PQ for growth in solid media.

For the Arabidopsis experiments, we chose oxidative stressors (and appropriate concentrations) based on the literature. We have added the respective citations to the text. These conditions were used for several published Arabidopsis microarray experiments that are used as a reference in the field. We avoided using PQ in Arabidopsis because PQ in plants primarily induces ROS production from the photosystems in chloroplasts. 

For yeast experiments in Fig. 3, we chose PQ because it is known to induce oxidative stress in mitochondria (doi: 10.1074/jbc.M708597200). In Fig. 3A, B, for the acute stress, we used a much higher concentration of PQ (2 mM) because we wanted to affect the WT to evaluate whether COX11 overexpression helps with the imposed stress. In Fig. 3C, for continuous stress, we used a much lower PQ concentration (0.5/0.2 mM PQ) to show that Δsod1Δcox11 mutants cannot grow on PQ concentrations, on which WT grew normally. And finally, we used different PQ concentrations for yeast growth on solid vs liquid media as yeast cells have different stress tolerances under these two growth conditions. We have added those clarifications to the results section of the text.

In Fig. 4 we used menadione, which is a general redox cycler and ROS inducer because the tCOX11 variants were expressed in the cytosol and not mitochondria which is the primary target for PQ. We have also added this clarification to the results section of the text.

4. The western blotting results in Fig 4B where the authors show the expression of Cox11 variants in yeast WT cells raises concerns because it is not clear as to why a band for endogenous WT Cox11 protein is not seen in all samples.

We do not see the endogenous COX11 protein band because we used total protein extract where native COX11 is present at a very low abundance and the polyclonal antibody was not sensitive enough to detect it. We have added this clarification to the results section of the text. 

5. The biological relevance of the results obtained using putative cytosol localized truncated Cox11 forms is questionable. WT Cox11 is an inner mitochondrial membrane protein and unlike some of the other IMS soluble proteins, it is neither dual localized nor can it shuttle between cytosol and mitochondria.

We purposefully chose to express the tCOX11 variants in the cytosol where they do not naturally reside, because we wanted to test whether COX11 proteins have antioxidative properties outside of their native environment where they also have complex protein-protein interactions, especially with the COX complex. This approach also allowed us to separate the COX11 antioxidative functions from their role in COX assembly. The tCOX11 proteins were expressed in the WT background, so different strains would not vary in respiratory competence. 

6. The use of DCFDA to measure cellular ROS should be complemented with MitoSOX to measure mitochondrial ROS in various truncation versions as well in mitochondrial localized Cox11.

In our experimental set-up, MitoSox unspecifically stained compartments other than mitochondria. An example image is shown below (MitoSox in red, Mitotracker in green, DAPI in blue). In addition, we are not sure of the benefit of measuring mitochondrial ROS in strains expressing truncated COX11 proteins that are NOT targeted to mitochondria. 

7. The data in Fig 3C right panel contradicts published literature on the sensitivity of yeast cox11∆ to oxidative stressors. See: “Khalimonchuk O, Bird A, Winge DR. Evidence for a pro-oxidant intermediate in the assembly of cytochrome oxidase. J Biol Chem. 2007 Jun 15;282(24):17442-9.”

For this, there are two possible explanations. Firstly, for this assay, we used a very low concentration of PQ which was probably not sufficient to limit the growth of Δcox11, Δcox7, and rho0 mutants. And secondly, the experimental setups used here and in the paper of Khalimonchuk et al., were quite different. We cultured yeast on continuous PQ stress, which could have potentially allowed cells to adapt. On the other hand in the Khalimonchuk et al. paper, yeast cells were exposed to short acute hydrogen peroxide stress and the survival was evaluated. 

Minor concerns:

1. The authors should provide information about the statistical tests used. (Ex: Fig 1B and C).

We have added that information to the Fig. 1 legend and methods. 

2. There are minor corrections required: change ScSOX11 to ScCOX11 in lines 286 and 287.

Done. Thank you for catching that. 

6. PLOS authors have the option to publish the peer review history of their article (what does this mean?). If published, this will include your full peer review and any attached files.

Do you want your identity to be public for this peer review? For information about this choice, including consent withdrawal, please see our Privacy Policy.

Reviewer #1: No

Reviewer #2: No

Reviewer #3: No

---

## [Decision Letter · Decision Letter 1]

3 Dec 2021

The mitochondrial copper chaperone COX11 has an additional role in cellular redox homeostasis

PONE-D-21-15368R1

Dear Dr. Radin,

We’re pleased to inform you that your manuscript has been judged scientifically suitable for publication and will be formally accepted for publication once it meets all outstanding technical requirements.

Kind regards,

Rodrigo Franco

Academic Editor

PLOS ONE

Additional Editor Comments (optional):

Reviewers' comments:

Reviewer's Responses to Questions

**Comments to the Author**

1. If the authors have adequately addressed your comments raised in a previous round of review and you feel that this manuscript is now acceptable for publication, you may indicate that here to bypass the “Comments to the Author” section, enter your conflict of interest statement in the “Confidential to Editor” section, and submit your "Accept" recommendation.

Reviewer #1: All comments have been addressed

Reviewer #2: All comments have been addressed

2. Is the manuscript technically sound, and do the data support the conclusions?

Reviewer #1: Yes

Reviewer #2: Yes

3. Has the statistical analysis been performed appropriately and rigorously? 

Reviewer #1: Yes

Reviewer #2: Yes

4. Have the authors made all data underlying the findings in their manuscript fully available?

Reviewer #1: Yes

Reviewer #2: Yes

5. Is the manuscript presented in an intelligible fashion and written in standard English?

Reviewer #1: Yes

Reviewer #2: Yes

6. Review Comments to the Author

Reviewer #1: The authors addressed several of the points raised on the previous versions. I am convinced that a further round of revision would not make this study stronger. I support publication of this study in its present form.

Reviewer #2: This manuscript presents COX11’s antioxidant function in addition to its known role in cytochrome oxidase assembly. In this revision, the authors adequately addressed this reviewer’s comments and suggestions. The data provide some valuable information about COX11’s functions and action mechanisms.

7. PLOS authors have the option to publish the peer review history of their article (what does this mean?). If published, this will include your full peer review and any attached files.

Reviewer #1: No

Reviewer #2: No

---

## [Editor Report · Acceptance letter]

10 Dec 2021

PONE-D-21-15368R1 

The mitochondrial copper chaperone COX11 has an additional role in cellular redox homeostasis 

Dear Dr. Radin:

I'm pleased to inform you that your manuscript has been deemed suitable for publication in PLOS ONE. Congratulations! Your manuscript is now with our production department. 

Kind regards, 

on behalf of

Dr. Rodrigo Franco 

Academic Editor

PLOS ONE